# Generalizing population RT-qPCR cycle threshold values-informed estimation of epidemiological dynamics: Impact of surveillance practices and pathogen variability

Yun Lin[1], James A. Hay[2], Yu Meng[1], Benjamin J. Cowling[1,3], Bingyi Yang[1]*

**1** WHO Collaborating Centre for Infectious Disease Epidemiology and Control, School of Public Health, Li Ka Shing Faculty of Medicine, The University of Hong Kong, Hong Kong Special Administrative Region, China, **2** Pandemic Sciences Institute, Nuffield Department of Medicine, University of Oxford, Oxford, United Kingdom, **3** Laboratory of Data Discovery for Health Limited, Hong Kong Science and Technology Park, Hong Kong Special Administrative Region, China

* byyang@connect.hku.hk

## Abstract

Population-level viral load distributions, measured by RT-qPCR or qPCR cycle threshold (Ct) values from surveillance testing, can be used to estimate the time-varying reproductive number ($R_t$) in real-time during COVID-19 outbreaks. However, it remains unclear whether this approach can be broadly applied to other pathogens, sources of virologic test data, or surveillance strategies beyond those specifically implemented during the COVID-19 pandemic in Hong Kong. We systematically evaluated the accuracy of Ct-based $R_t$ estimates using simulated epidemics under different surveillance testing systems and pathogen viral kinetics. Using area under the ROC curve (AUC) to assess accuracy in detecting epidemic growth or decline, we found that case ascertainment rates minimally impacted estimation accuracy, except when detection was heavily biased towards severe patients (AUC: 0.64, 95% CIs: 0.59 - 0.71) or during prolonged waves with stable $R_t$ near one (AUC: 0.54, 0.48 - 0.64), compared to stable detection patterns over time (AUC 0.76, 0.66 - 0.82). By comparing model accuracies across different viral shedding patterns and by parameterizing our model using data from six respiratory pathogens, we found that model performance largely depends on a monotonic viral shedding trajectory following case detection. A pathogen that lacks such shedding pattern – for example, those with a viral peak after onset – exhibited lower accuracy (AUC: 0.58, 0.49 - 0.65). Overall, our findings demonstrate that Ct-based $R_t$ estimation methods are generally accurate across diverse surveillance conditions and pathogen shedding patterns, supporting their practical use as a supplementary tool for timely transmission monitoring while highlighting limitations that warrant further consideration.

**Data availability statement:** Parameters for simulations can be found in Methods and Supplementary Materials. All simulation data generated in this study and all codes for analyses are available at the GitHub repository (https://github.com/vanialin/Ct_Rt_generic).

**Funding:** This study was supported by the Health and Medical Research Fund from the Health Bureau of the Government of the Hong Kong Special Administrative Region (grant no. 22210552, B.Y.) and by a grant from the Research Grants Council of the Hong Kong Special Administrative Region, China (Project No. T11-705/21-N, B.J.C.) The funders had no role in study design, data collection and analysis, decision to publish, or preparation of the manuscript.

**Competing interests:** I have read the journal's policy and the authors of this manuscript have the following competing interests: BJC consults for AstraZeneca, Fosun Pharma, GSK, Haleon, Moderna, Novavax, Pfizer, Roche, and Sanofi Pasteur. The authors report no other potential conflicts of interest.

## Author summary

Population viral load distributions, often approximated by cycle threshold (Ct) values from RT-qPCR testing, have proven valuable for real-time estimation of transmission rates, enhancing situational awareness during the COVID-19 pandemic. However, a comprehensive framework for applying Ct-based methods in other epidemiological contexts, such as varying levels of surveillance coverage or different circulating pathogens/variants, has yet to be developed. In this study, we evaluated the strengths and limitations of Ct-based epidemic surveillance approaches by simulating a range of scenarios with diverse surveillance coverage reflecting real-life outbreaks and carefully calibrating pathogen viral kinetics using real-world parameters. Our findings demonstrate that Ct-based $R_t$ estimates are generally accurate across a range of surveillance and pathogen conditions, supporting their utility as a supplementary tool for timely epidemic monitoring while also highlighting limitations for consideration in future applications.

## Introduction

Monitoring infectious disease transmission is important for implementing effective public health measures and timely assessment of the effectiveness of interventions. The time-varying effective reproductive number ($R_t$), defined as the expected number of secondary infections per infectious individual, has been widely used to monitor community transmission, especially during the COVID-19 pandemic [1–3]. Conventionally, $R_t$ estimates rely on statistical models applied to time series of reported case counts (i.e., the incidence-based method), which suffers unavoidable delays due to the time between infection, symptom development, diagnosis and reporting [4,5].

To provide an alternative method for real-time $R_t$ estimation, inference models have integrated population-level viral load data measured through cycle threshold (Ct) values from reverse-transcription quantitative polymerase chain reaction (RT-qPCR) [6,7]. This approach, validated in simulations using random Ct samples [6] and applied in Hong Kong's epidemic waves with symptom and contact-tracing-based surveillance [7], leverages Ct value data collected on a given day to provide earlier insights into transmission dynamics. However, factors such as changes in surveillance practices, including the proportion of detected cases, and testing capacity, reasons for selecting individuals for testing, and variation in viral kinetics across different pathogens and variants can impact the distribution of viral loads measured in the population, limiting the method's generalizability.

Recent research highlights the value of Ct-based methods in nowcasting and forecasting transmission across regions and SARS-CoV-2 variants including Alpha, Delta and Omicron [8–10]. Evidence from Hong Kong also demonstrated the method's

accuracy during Omicron waves while emphasizing the impact of significant detection delays on method performance [11]. While these findings support the method's application during epidemic waves with diverse variants and changing population immunity, a systematic evaluation of the method in the context of varying surveillance efforts, or its adaptability to other viruses, remains limited.

Considering the potential future risks of outbreaks posed by emerging or re-emerging pathogens [12], it is crucial to examine and refine new real-time surveillance metrics, including Ct-based methods, as supplementary tools that could provide additional and timely insights to traditional incidence-based $R_t$ for future applications. Here, we compared the performance of Ct-based $R_t$ estimation methods in simulated scenarios featuring different detection and testing coverage, epidemic characteristics, and pathogen outbreaks with varied viral kinetics, aiming to build a generic framework for incorporating population viral load into $R_t$ nowcasting for future application.

## Results

### Overview of scenarios and procedures for comparative analysis

The Ct-based method applies a statistical model trained to relate incidence-based $R_t$ to daily population Ct value statistics (i.e., mean and skewness), allowing real-time $R_t$ estimation from observed Ct values (Fig 1A). Both surveillance coverage (e.g., the proportion of cases detected and tested by RT-qPCR) and viral shedding kinetics can affect daily Ct value distributions and therefore also the $R_t$ estimation (Fig 1A). To systematically assess these factors, we simulated two consecutive epidemic waves (Table A in S1 Text; see **Methods**), using the first wave as the model training period and the second wave to evaluate estimation accuracies (Fig 1).

In the first set of simulations, we simulated epidemics of a SARS-CoV-2 ancestral strain-like pathogen while varying surveillance practices. Specifically, we adjusted the proportion of cases detected and the proportion of detected cases that were tested by RT-qPCR (Fig 1B). Scenarios included reducing the fraction of detected cases tested from 100% to 30% (scenarios 1–4), as well as epidemic plateaus that arose either from limited detection capacity (i.e., only able to detect a limited number of cases per day; scenario 5) or from a stabilized $R_t$ near one (scenario 6). We also modelled detection bias by varying the proportion of detected severe cases relative to non-severe cases, as severe cases typically have longer shedding durations and lower detected Ct values [13], and included scenarios both with and without additional detection delays for these cases (scenarios 7–14) (Fig 1B; Table B in S1 Text).

The second set of simulations focused on the impact of viral shedding kinetics, holding surveillance constant but introducing pathogens or variants with different shedding kinetics (Fig 1C). We compiled four parameter sets with distinct timings of peak viral load relative to illness onset (Fig 1C; see **Methods**). Viral shedding parameters for six real-world pathogens, including different variants of SARS-CoV-2 (ancestral strain, Alpha, Delta and Omicron variants), SARS-CoV-1 and influenza A (Table C in S1 Text; see **Methods**), were applied to further elucidate the key pathogen characteristics necessary for accurate estimation.

We simulated transmission rates using a susceptible-exposed-infectious-recovered (SEIR) model as the simulation truth, applying varied surveillance and pathogen parameters to generate daily case numbers and associated Ct values for each scenario. Detected cases were used to estimate incidence-based $R_t$, while population-level Ct distributions (summarized by mean and skewness by sampling dates) were used to estimate Ct-based $R_t$ following our previous method [7] (see **Methods**). The model was trained using data from three weeks before and after the peak case date during the first epidemic wave (day 0–109 after the initial outbreak; see S1 Text), and tested on the subsequent testing period (110 days after initial outbreak). We primarily compared Ct-based $R_t$ estimates with the simulation truth using the area under the receiver operator characteristic curve (AUC; see **Methods**), which evaluates the model's ability to distinguish between high ($R_t \geq 1$) and low ($R_t < 1$) transmission rates. Additional evaluation metrics, including Spearman's correlation coefficient $\rho$ and mean absolute percentage error (MAPE), were also used (see S1 Text).

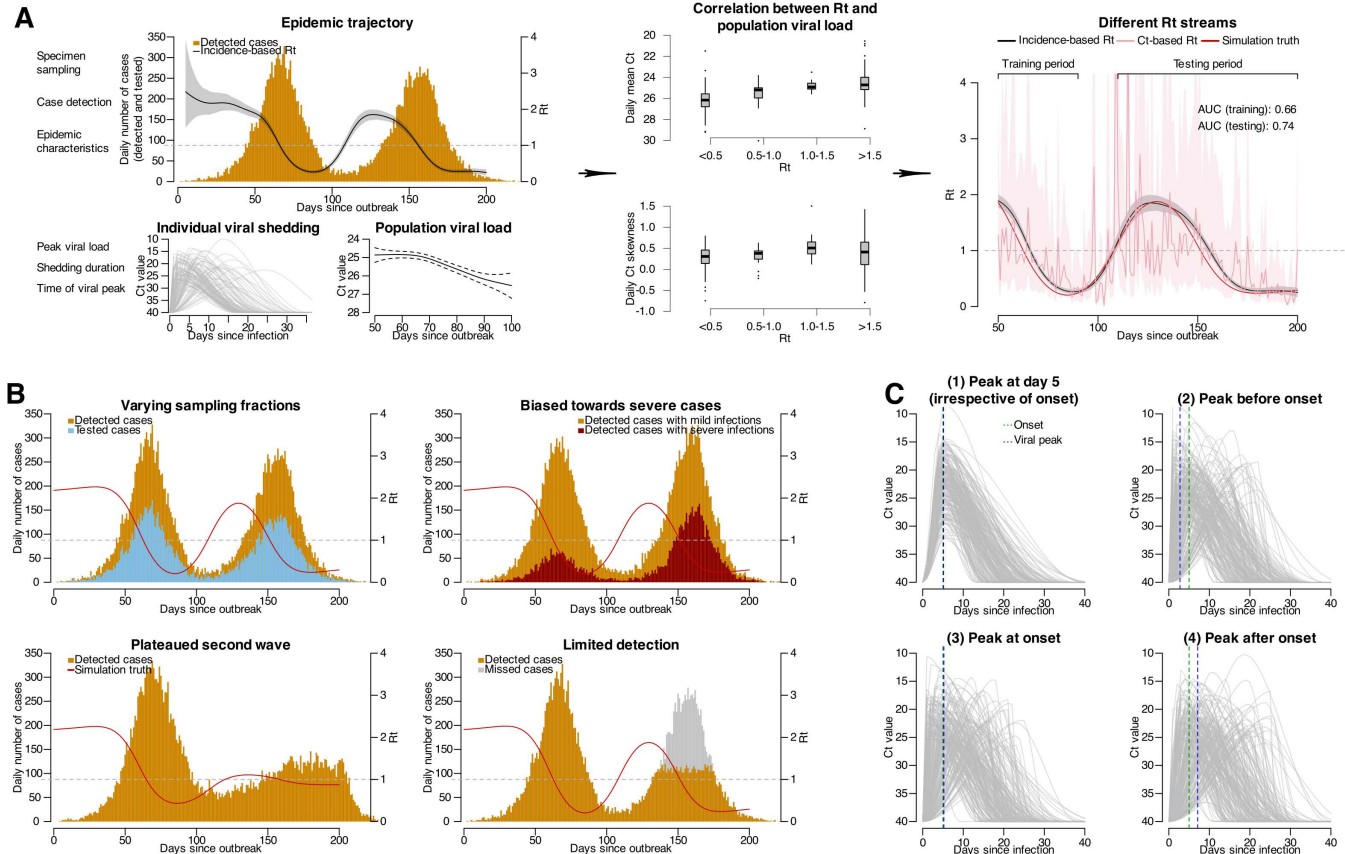

**Fig 1. Overview of data flows used to generate Ct-based Rt and factors potentially affecting distributions of population Ct values.** (A) the mechanism of Ct-based Rt illustrated by the relation between transmission dynamics (epidemic trajectory; top left panel) and population viral load distributions (bottom left panels). Correlations between population Ct distributions as indicated by daily mean and skewness are shown in middle panels, with boxes representing the estimated median and interquartile range (IQR) of daily Ct distributions under each interval of the simulation truth Rt, the lower and upper whiskers representing the minimum and maximum estimates and dots for outliers. Temporal transmission dynamics as indicated by Rt estimated using different data are shown in the right panel. Black line and gray shaded area indicate median and 95% credible intervals for incidence-based Rt, pink line and shaded area indicate median and 95% confidence intervals for Ct-based $R_t$, while red line indicate simulation truth (i.e., the real transmission rates under simulation). Area under the receiver operator characteristics curve (AUC) is used to evaluate estimation accuracy, with the AUC for Ct-based Rt compared to the simulation truth over training and testing periods denoted in top-right of the right panel. (B) epidemic trajectories under different detection modes, as examples of varying surveillance sensitivity. Bars show the number of cases, with different colors indicating different case types as denoted in corresponding panels. Red line indicates the trend for simulation truth. (C) viral shedding trajectories for four pathogens (pathogen 1-4) with varying timings of viral peak in relation to illness onset, as the other affecting factor (i.e., pathogen viral kinetics) for model performance. In each panel, 200 infected individuals are randomly selected to demonstrate the shedding trajectory for each pathogen, and each gray line indicates the shedding trajectory of an infected individual. Green and blue vertical dashed lines indicate the median time of onset and viral peak among the 200 selected individuals infected with each of the four pathogens.

## Impact of varying surveillance coverage

The Ct-based $R_t$ estimation achieved an AUC of 0.76 (95% confidence intervals (CIs): 0.66 - 0.82) during the testing period in scenario 1, which modelled epidemics caused by a SARS-CoV-2 ancestral strain-like pathogen with stable detection and 100% of cases tested with RT-qPCR. Estimation accuracy remained moderate to high even when only 30% of detected cases reported Ct values (scenario 4; AUC = 0.66, 95% CIs: 0.58 – 0.75), or when daily case detection was limited (scenario 5; AUC = 0.74, 95 CIs: 0.65 – 0.81) (Figs 2 and B in S1 Text). Moderate detection bias towards severe cases did not significantly affect accuracy (scenarios 7–9; AUC ranging from 0.75 to 0.76), but delayed detection of severe

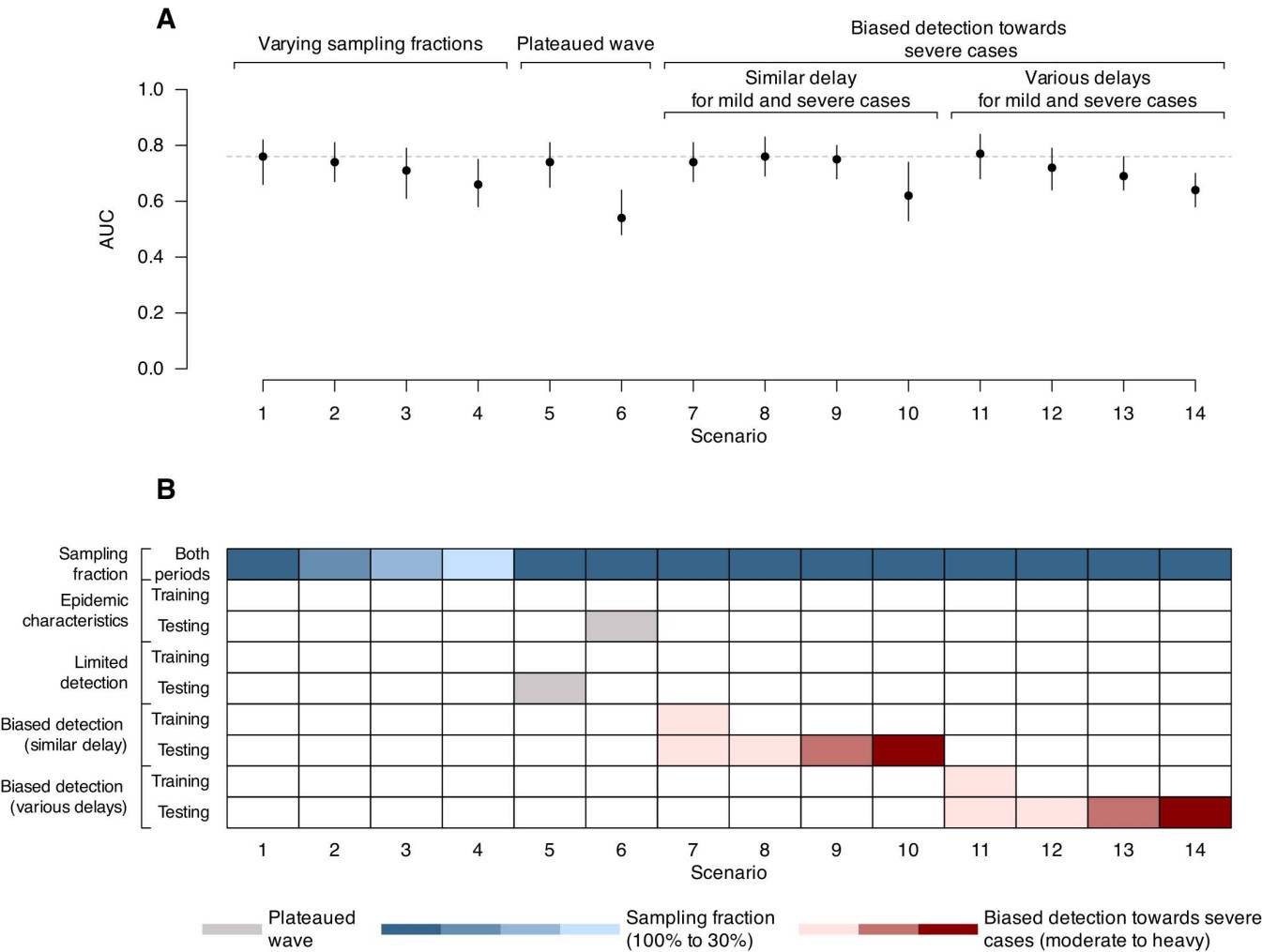

**Fig 2. Estimation accuracy for Ct-based Rt during the testing period in different scenarios of surveillance sensitivity.** (A) estimation accuracy for Ct-based Rt as compared to the simulation truth and evaluated by AUC over the testing period in each scenario. Points and vertical lines show median and interval estimates of the AUC over 100 times of bootstrapping (see Methods), with the interval estimates taken as the 2.5 and 97.5 percentiles of all 100 estimated AUC values during bootstrapping. (B) brief summary of each scenario in terms of surveillance-related factors that can affect model performance, as illustrated in Fig 1A. More detailed description of each scenario can be found in Table B in S1 Text.

cases reduced performance (scenarios 12–14 vs. scenarios 8–10) (Figs 2 and B in S1 Text). With strong detection bias (i.e., testing only severe cases), accuracy dropped to an AUC of 0.62 – 0.64 (scenarios 10 and 14), though it remained moderately accurate (Figs 2 and B in S1 Text).

Estimation accuracy varied according to the underlying cause of an observed flattened epidemic curve. When the plateau resulted from limited detection (scenario 5), the Ct-based $R_t$ estimate accuracy remained high (AUC = 0.74, 95% CIs: 0.60 – 0.82). In contrast, during a genuine low transmission period (scenario 6), accuracy declined significantly (AUC = 0.54, 95% CI: 0.48 – 0.64) (Fig 2). This decrease was supported by lower Spearman's correlation and higher MAPE (scenario 6 in Fig C in S1 Text; see S1 Text). An overview of the temporal trend of all $R_t$ time series can be found in Fig D in S1 Text. Of note, lower estimation accuracy was typically associated with less variable population Ct values over time, such as observed in low transmission (panel B of Fig E in S1 Text) and heavily biased scenarios (panel D of Fig E in S1 Text).

Overall, the method performed best during epidemic growth, with both stable (scenario 1) and limited detection (scenario 5) conditions outperforming low transmission scenario (scenario 6) (Fig F in S1 Text). Although performance improved with increased Ct sample sizes, benefits became marginal beyond 100 samples (Fig G in S1 Text; see S1 Text).

### Impact of distinct viral shedding kinetics

We compared the estimation accuracies of four viral shedding kinetics models that differ solely in the timing of their viral load peak relative to illness onset, while keeping peak viral loads and shedding durations similar. The model with an earlier peak than illness onset (pathogen 2) achieved high accuracy (AUC = 0.85, 95% CIs: 0.78 − 0.90), whereas the model with a delayed viral peak than illness onset (pathogen 4) showed much lower accuracy (AUC = 0.58, 95% CIs: 0.49 − 0.65; Fig 3B). The performance differences are consistent with the relationship between population Ct values and epidemic progression, as pathogens with greater variations in population Ct values during epidemic growth and declines and therefore a clear and linear relationship between population Ct values and epidemic progression tend to yield higher model accuracy (Figs 3 and H in S1 Text).

The varying performance across viral shedding kinetics can be attributed to how well exposure time can be inferred from viral loads. With fixed shedding durations, an earlier viral peak allows a monotonic shedding trajectory for detection under symptom-based surveillance, resulting in a clear relationship between time-since-infection and viral loads that helps distinguish recent versus older infections. Consequently, population Ct values differ significantly between epidemic growth and decline phases, with mean Ct values ranging from 26.1 to 28.7 and skewness ranging from 0.5 to 0.3 at the epidemic peak versus the decreasing phase (Fig 3D). Conversely, a delayed viral peak leads to non-monotonic trajectories, increasing the likelihood that recent infections are detected before reaching peak viral load. Consequently, mean Ct values remained around 26 and skewness at 0.5 across epidemic phases, thereby diminishing the signal of transmission dynamics (Fig 3F).

### Application potential for real-world pathogens

We further demonstrated the method's potential across various real-world pathogens and variants. For SARS-CoV-2, the Alpha (AUC = 0.81, 95% CIs: 0.76 − 0.88) and Delta (AUC = 0.82, 95% CIs: 0.76 − 0.90) variants showed higher accuracy than the ancestral strain, likely due to their relatively earlier and higher viral peaks at or before symptom onset. In contrast, Omicron infections initially yielded less favorable performance (AUC = 0.75, 95% CIs: 0.65 − 0.82), likely due to a later viral peak and non-monotonic shedding trajectories post- detection (Figs 4 and I in S1 Text). However, delaying Ct testing by 2–4 days to capture the monotonic phase improved the performance (AUC = 0.84, 95% CIs: 0.78 − 0.89) (Fig J in S1 Text; see S1 Text).

For SARS-CoV-1, a reversed relationship between Ct and $R_t$ was observed due to distinct shedding dynamics, yet high accuracy was achieved (AUC = 0.88, 95% CIs: 0.82 − 0.92), possibly due to a later viral peak that provided a stable time-dependent signal for inferring exposure (Figs 4B and I in S1 Text). Conversely, influenza A exhibited reduced performance (AUC = 0.61, 95% CIs: 0.53 − 0.69) because its very short shedding duration resulted in consistently low population viral load (Figs 4 and I in S1 Text).

### Discussion

In this study we examined the potential of Ct-based $R_t$ estimation methods across different surveillance practices and viral shedding kinetics. Our findings indicate that larger temporal variations in population Ct values yield clearer epidemic signals and more accurate Ct-based $R_t$ estimates. These variations are driven more by pathogen shedding kinetics than by surveillance practices, except in cases of heavily biased detection or stagnant epidemics conditions. A monotonic shedding trajectory is crucial for effectively distinguishing recent from older infections, enhancing the model's ability to capture epidemic phases. By demonstrating that Ct-based $R_t$ estimation methods remain accurate across diverse surveillance

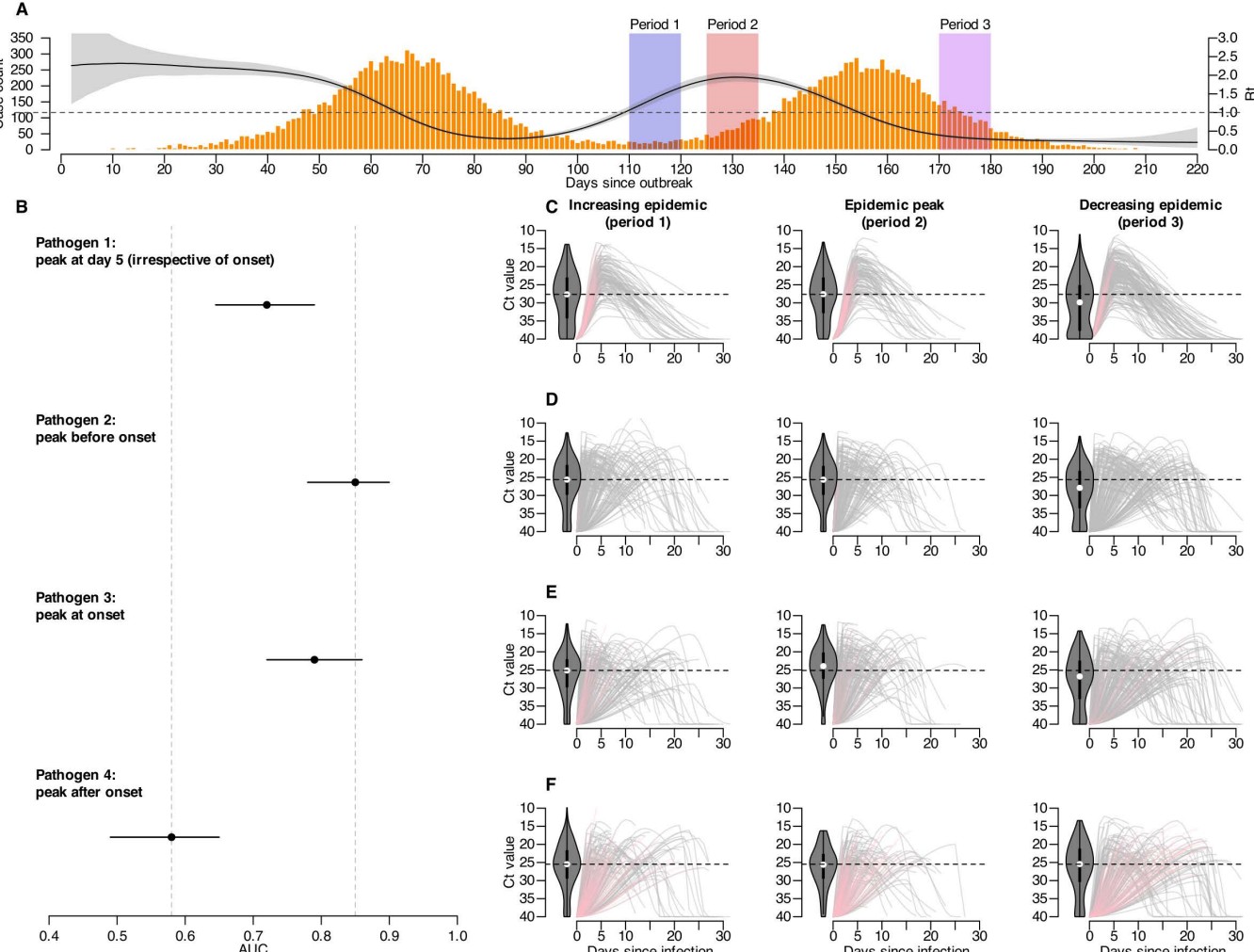

**Fig 3. Impact of pathogen shedding characteristics on estimation accuracy.** (A) Summary of epidemic waves in the scenario of stable detection. Black solid line and gray shaded area represents the median and 95% credible intervals of incidence-based Rt, orange bars show daily number of detected cases, and shaded areas with different colors indicate different epidemic periods (increasing epidemic as blue-shaded period 1, epidemic peak as red-shaded period 2, and decreasing epidemic as purple-shaded period 3). (B) estimation accuracies of Ct-based Rt over testing periods for pathogen 1-4 corresponding to examples shown in Fig 1C. Dots and horizontal lines represent median and interval estimates of the AUC over 100 bootstrapped samples(see Methods), with the interval estimates taken as the 2.5 and 97.5 percentiles of all 100 estimated AUC values during bootstrapping. Gray dashed lines in the background mark the minimum and maximum median AUC across the four pathogen scenarios serving as visual reference points for easier comparison. (C-F) trajectories of viral shedding until the time of detection for 200 randomly selected individuals infected with pathogen 1-4 corresponding to examples shown in Fig 1C, and in different epidemic periods as indicated in panel A (left panels: increasing epidemic, period 1; middle panels: epidemic peak, period 2; right panels: decreasing epidemic, period 3). Each line represents the viral shedding trajectory for each individual, and a gray line indicates the trajectory for an individual that is detected after viral peak, while a pink line indicates that for an individual detected before viral peak. The violin plot represents the distribution of the detected Ct values for the 200 randomly selected individuals in each trajectory panel, and the horizontal dashed line represents the median Ct value at detection for individuals infected with each pathogen during the increasing epidemic as a reference for comparing the Ct distributions for each pathogen during different epidemic periods. (C) Shedding trajectories until detection for individuals infected with pathogen 1 whose viral loads peak at day 5 after infection (irrelevant to the timing of symptom onset). (D) Shedding trajectories until detection for individuals infected with pathogen 2 whose viral loads peak 2 days before symptom onset. (E) Shedding trajectories until detection for individuals infected with pathogen 3 whose viral loads peak at symptom onset. (F) Shedding trajectories until detection for individuals infected with pathogen 3 whose viral loads peak 3 days after symptom onset.

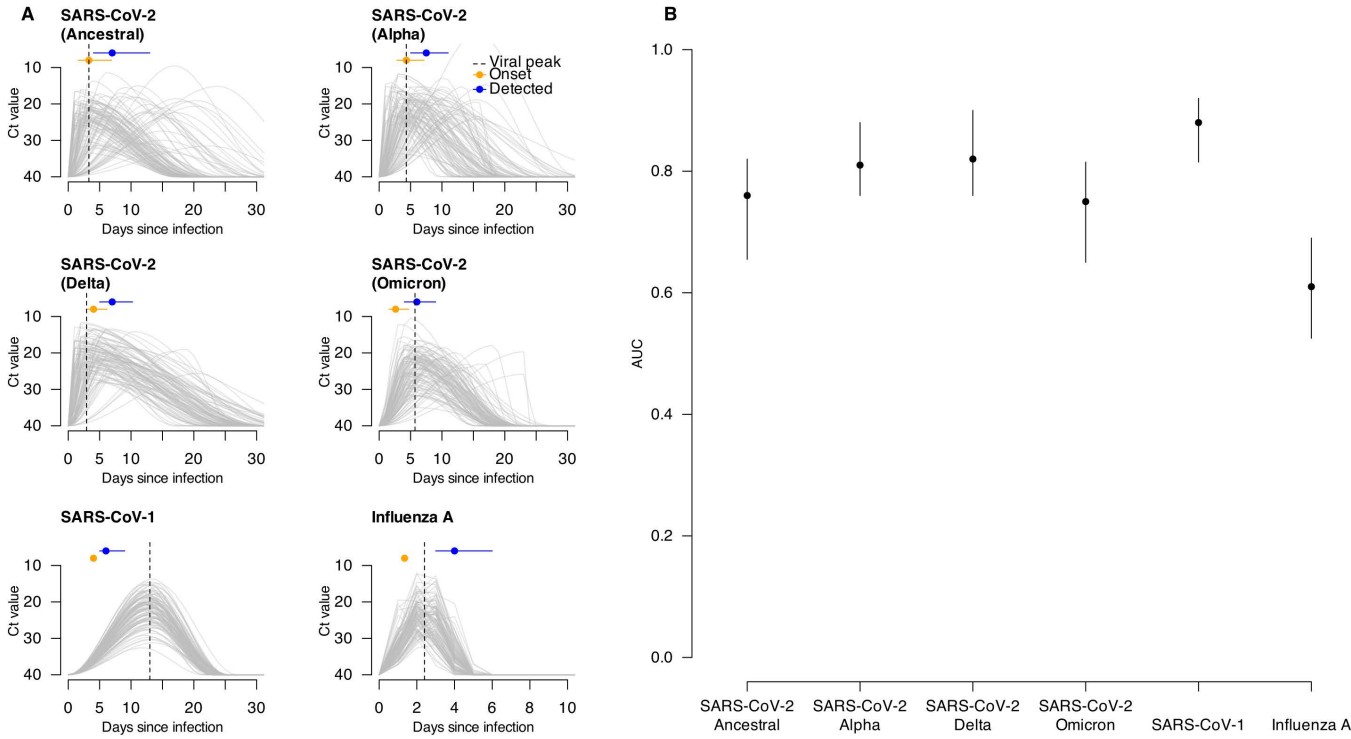

**Fig 4. Viral shedding trajectories for six real-world pathogens and estimation accuracies of Ct-based Rt over testing periods for these pathogens.** (A) viral shedding trajectories of 100 randomly selected individuals infected with each of the six real-world pathogens. Each gray line indicates the shedding trajectory of an infected individual. The dashed vertical line indicates the median time of viral peak for the 100 individuals. Orange dots and horizontal lines represent the median and IQR of the onset time for these individuals (Table C in S1 Text), while blue dots and lines represent the median and IQR of their detection time (Table A in S1 Text) (see Methods). (B) estimation accuracies of Ct-based Rt during the testing period for six real-world pathogens. Dots and vertical lines show median and interval estimates of the AUC over 100 times of bootstrapping (see Methods), with the interval estimates taken as the 2.5 and 97.5 percentiles of all 100 estimated AUC values during bootstrapping.

conditions and pathogen shedding patterns, our framework supports leveraging routinely collected RT-qPCR Ct data, which is often overlooked in surveillance, as a supplementary real-time metric alongside traditional incidence-based $R_t$. Such incorporation of Ct-based $R_t$ can offer timely and complementary insights into epidemic trends, which can be particularly useful for early outbreak surveillance in the future.

Our evaluation across varying surveillance practices identified key factors affecting model performance. As the method relies on the distribution of Ct values rather than absolute case counts, even limited or moderately biased detection can provide informative signals of epidemic changes, as observed in surveillance scenarios 5 and during the Omicron waves in Hong Kong when changes in testing and case ascertainment likely decreased the accuracy of incidence-based $R_t$ estimates [7,11]. However, dramatic shifts, such as a sudden bias in detecting only severe cases, which typically have longer shedding durations and lower Ct values, or delays in detecting these cases, can distort the learned Ct-$R_t$ relationship and reduce estimation accuracy, as observed in some of our simulated surveillance scenarios and during epidemic periods when sample representativeness changed or case numbers plateaued [11]. Our simulations provide possible explanations in relation to these problems, helping to understand the strengths and caveats of the method for practical application.

We demonstrated that the relationship between population Ct values and $R_t$ estimates remain robust when average exposure time can be accurately inferred from viral load data. Pathogens with earlier and higher viral peaks, coupled with moderate shedding durations, exhibit greater temporal variability in Ct values from recent and older infections, thereby

enhancing the model performance. In contrast, extremely short shedding duration (as seen in our influenza A example [14]) or non-monotonic shedding trajectories (as with the Omicron example [15]) limit the ability to estimate time-since-infection, potentially compromising accuracy. Notably, our findings suggested that adjusting the Ct testing window to capture the Omicron's monotonic shedding phase could improve the performance.

While our study focused on applying this methods to human respiratory viruses, the framework could be extended to other pathogens where viral loads correlate with exposure time. For instance, mpox has an incubation period of around 5 days [16], with viral shedding peaking within a week after symptom onset and PCR positivity lasting up to 3 weeks [17,18]. Since detection typically occurs 3–5 days after onset [17] during the viral proliferation phase, Ct-based estimates may be as accurate as for SARS-CoV-1. In H5N1 outbreaks, prolonged viral shedding in birds [19] could enable reliable $R_t$ estimation using Ct values, despite challenges such as species-specific variability, asymptomatic shedding [19–21] and the need for additional surveillance data, particularly systematic Ct value collection from at-risk or infected poultry and exposed individuals. Although we did not explore these pathogens due to differences in transmission routes and surveillance strategies, our findings imply promising directions for enhancing future surveillance.

Although we have analyzed various surveillance practices and circulating pathogens, our assumptions about detection mechanisms, including who is tested and when, may not fully capture real-world complexities. Additionally, factors such as co-circulating virus subtypes and heterogeneous immune backgrounds can further affect the representativeness of observed Ct distributions and the accuracy of resulting $R_t$ estimates. Shifts in dominant variants across epidemic waves, such as Omicron in Hong Kong [22], may further complicate predictions. Reported viral shedding parameters also differ between studies; for instance, one study [23] found longer influenza A shedding durations than those used in our analysis [14,24]. However, we believe such heterogeneity has minimal impact on the method's applicability, as our simulations and observations from epidemic waves in Hong Kong [11] demonstrate that the method remains accurate even under varying severity profiles, viral variants, population immunity, and surveillance intensities.

In addition, estimating $R_t$ from daily Ct distributions produces more variable estimates than incidence-based $R_t$, which typically applies a 7-day or 14-day smoothing window [3]. Previous work found that simple statistical smoothing of Ct values did not improve performance and sometimes reduced accuracy, likely because it disrupted the temporal relationship between Ct distributions and $R_t$[11]. Future methodological improvements that reduce daily fluctuations in population Ct values while preserving accurate and timely nowcasting could improve the utility of this approach. While our analysis focused on symptomatic individuals typically detected through targeted and syndromic surveillance, future research should also explore incorporating Ct values from asymptomatic cases in settings with more comprehensive and universal testing. Additionally, expanding analyses or validating this method under more complicated and varying surveillance conditions in simulations or real-world settings will provide further insights and point to important directions for future work.

While our current approach uses incidence-based $R_t$ estimates to calibrate the Ct-based model, future work could develop fully independent Ct-based models that do not require case incidence data for training. Because incidence-based $R_t$ estimation depends on delay distributions and incubation periods that are often uncertain and may introduce bias, Ct-based methods developed independently of incidence data could help overcome these limitations. Further development of these models could also better account for complexities such as non-monotonic shedding patterns, thereby further enhancing the robustness and applicability of Ct-based surveillance methods.

To summarize, our modelling study extends the use of Ct-based $R_t$ estimation methods in symptom-based surveillance settings to scenarios with different surveillance practices (e.g., variations in case ascertainment, virologic testing and severity biases), as well as different pathogen viral kinetics. This generic framework identifies key factors affecting the model performance and highlights the value of incorporating viral load data to monitor transmission dynamics and enhance preparedness and response strategies against various infectious threats.

## Methods

### Simulation of epidemic waves, detected individuals, and viral loads

We used an SEIR model to simulate two consecutive epidemic waves (see S1 Text). Two consecutive epidemic waves with clear increasing and decreasing phases were simulated, while another scenario (scenario 6) featured a typical first wave and a second wave with $R_t$ stable around one using a different set of parameters (Table A in S1 Text). We focused on symptom-based surveillance, where individuals are detected only if they develop symptoms after infection. Therefore, detection dates occur on or after symptom onset with a detection delay modelled by a gamma distribution (mean: 4.20, SD: 3.14), and these detection dates are used as testing dates for Ct values unless otherwise specified.

For the typical scenario of stable case detection similar to the first four COVID-19 waves in Hong Kong [25] (i.e., scenario 1), daily detected symptomatic cases were simulated by drawing from a binomial distribution: *Binomial*($N = number\ of\ new\ infection,\ p = 0.25$). To reflect small daily fluctuations in detection probability, noise (e.g., ±0.5%) was added around the mean probability of 25%. Similarly, we synthesized different series of detection and testing probabilities to represent various surveillance modes that may occur in reality, including limited viral testing as well as biased case detection [26,27]. We explored the impact of testing only small fractions of detected cases (i.e., limited testing capacity) by varying the proportion of detected cases that were tested for Ct values from 100% to 80%, 50% and 30% respectively (scenarios 1–4; Table B in S1 Text). To recover plateaued epidemic waves caused by different reasons, we constructed a scenario of a second wave plateaued due to limited detection capacity (scenario 5) by restricting the number of detected cases at around 100 during the wave, while we also simulated the other possible scenario with genuinely low transmission (scenario 6) by updating the $R_t$ to be stably around one (Tables A and B in S1 Text).

During the Omicron waves in Hong Kong, the health system was challenged by the massive number of infected individuals, and severe patients were more likely to be captured and included in the hospital systems, causing certain biases in the surveillance [22]. To explore the impact of biased detection similar to this, we considered different case severity profiles and assigned corresponding detection probabilities. Specifically, we assumed 15%, 3% and 2% of cases would develop severe, critical or fatal illness respectively [28], which would lengthen their shedding durations (see S1 Text) and increase detection probabilities. While symptomatic cases generally have a 25% detection possibility, severe cases would have a 37.5% detection probability (1.5 times the baseline) and mild cases a 20% detection probability (80% of the baseline) in scenarios of biased detection. By varying detection rates for severe cases, the observed severity profile among detected cases would change over time, independent of any actual changes in the pathogen's genuine severity profile. We constructed scenarios with varying intensities and durations of biased detection (scenarios 7–10), and included scenarios with further delayed detection for severe cases (gamma distribution of mean = 6.95, SD = 4.12 compared to the original mean of 4.20) in scenarios 11–14 for comparison (Tables A and B in S1 Text).

For individual viral loads, we simulated viral load trajectories accounting for variation in observed Ct values each day since infection for infected symptomatic cases using previous method [7,29] (see S1 Text; Table A in S1 Text). Each infected individual had their Ct values per day since infection and, if detected, would have their corresponding sampled Ct values as the Ct value falling on day $k$ post infection based on their individual Ct trajectories, with $k$ being the time interval between their dates of infection and detection/sampling.

### Viral shedding parameters for different pathogens

To explore the impact of pathogen properties, we compiled four sets of viral shedding kinetics with different timings of viral peak relative to illness onset. Pathogen 1 has a viral peak at day 5 after infection (irrespective of onset), while pathogens 2, 3, and 4 have viral peaks 2 days before, at and 3 days after symptom onset, respectively (Fig 1C). We updated the distribution of viral shedding durations to align with the time of infection rather than illness onset to allow more flexible comparisons (see S1 Text). In symptom-based surveillance settings where individuals are detected and tested only after

developing symptoms, this adjustment enabled us to explore variations in the magnitude of detectable viral loads from different pathogens. We used the detection probability (25%) and detection delay distribution (gamma distribution with shape of 1.83 and rate of 0.43) from scenario 1 to generate the daily number of infected and detected cases, applying this to all four pathogens. Consequently, only the Ct values of detected cases and thus the population Ct values would vary across pathogen scenarios.

In addition to the four hypothesized time relations between viral peak and illness onset, we also used shedding parameters from the literature for variants of SARS-CoV-2, SARS-CoV-1 and influenza A as real-world examples to demonstrate the applicability of the Ct-based method (see S1 Text). Similarly, we used the detection probability (25%) and detection delay distribution (gamma distribution with shape of 1.83 and rate of 0.43) from scenario 1 to get daily number of infected and detected cases; however, due to differences in incubation periods and viral shedding parameters, both daily case counts and population Ct values will vary per pathogen scenario. We additionally explored different Ct testing windows for the Omicron scenario by extending Ct testing 2–4 days after case detection (see S1 Text).

### Incidence-based and Ct-based $R_t$

Incidence-based $R_t$ was estimated using the R package EpiNow2 [30] (see S1 Text), while daily distribution of population Ct (by sampling date $t$) estimated by mean ($\overline{x}_t$) and skewness ($b_t$) was used to generate Ct-based $R_t$ following our previous method [7]:

$$\ln(R_t) = \gamma_0 + \gamma_{\overline{x}}\overline{x}_t + \gamma_b b_t$$

Where $\ln(R_t)$ refers to natural log-transformed incidence-based $R_t$. To establish a generic training period applicable to all scenarios, we systematically evaluated different candidate training periods during the first simulated wave. Specifically, we varied the start dates and durations of these potential training periods, fitted regression models to each, and compared their adjusted R-squared values. For each scenario, the training period that produced the highest adjusted R-squared was selected as the best-performing period (see S1 Text). The generic characteristics of the best-performing training periods were then summarized and such training periods were applied across all scenarios to estimate the Ct-based $R_t$. The predicted Ct-based $R_t$ from days 110 onward (i.e., the testing period) were compared with $R_t$ from the SEIR model (i.e., simulation truth).

The area under the receiver operator characteristic curve (AUC) was used as the primary metric for assessing estimation accuracy, measuring the possibility that the Ct-estimated $R_t$ values correctly align with the simulation truth (i.e., both below or above 1). To investigate the uncertainty in sampling Ct values and consequently the accuracy of Ct-based $R_t$ estimates, we repeated each scenario 100 times using bootstrapping and calculated the AUC between the estimated Ct-based $R_t$ and the simulation truth in each iteration (see S1 Text). The estimation accuracy for each scenario was summarized by the median, 2.5% and 97.5% quantiles of the AUC across the 100 bootstrapping iterations.

All statistical analyses were conducted in R version 4.3.2 (R Development Core Team, 2023).

## Supporting information

**S1 Text. Supplementary Methods, Tables and Figures.**
(DOCX)

## Author contributions

**Conceptualization:** Yun Lin, Bingyi Yang.

**Data curation:** Yun Lin.

**Formal analysis:** Yun Lin.

**Funding acquisition:** Benjamin J Cowling, Bingyi Yang.

**Investigation:** Yun Lin, James A. Hay, Yu Meng.

**Methodology:** Yun Lin, James A. Hay, Bingyi Yang.

**Project administration:** Bingyi Yang.

**Resources:** Benjamin J Cowling, Bingyi Yang.

**Supervision:** Bingyi Yang.

**Validation:** James A. Hay.

**Visualization:** Yun Lin.

**Writing – original draft:** Yun Lin.

**Writing – review & editing:** James A. Hay, Yu Meng, Benjamin J Cowling, Bingyi Yang.

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
