## [Decision Letter · Decision Letter 0]

3 Jul 2025

Generalizing population RT-qPCR cycle threshold values-informed estimation of epidemiological dynamics: Impact of surveillance practices and pathogen variability

PLOS Computational Biology

Dear Dr. Yang,

Thank you for submitting your manuscript to PLOS Computational Biology. After careful consideration, we feel that it has merit but does not fully meet PLOS Computational Biology's publication criteria as it currently stands. Therefore, we invite you to submit a revised version of the manuscript that addresses the points raised during the review process.

Please submit your revised manuscript within 60 days Sep 02 2025 11:59PM. If you will need more time than this to complete your revisions, please reply to this message or contact the journal office at ploscompbiol@plos.org. Please include the following items when submitting your revised manuscript:

We look forward to receiving your revised manuscript.

Kind regards,

James M McCaw, PhD

Academic Editor

PLOS Computational Biology

Benjamin Althouse

Section Editor

PLOS Computational Biology

**Journal Requirements:**
**Reviewers' comments:**

Reviewer's Responses to Questions

**Comments to the Authors:**

Reviewer #1: Overall, this is a good manuscript that could be published in PLoS Computational Biology, following some additional analyses.

My main concern is that the study does not currently motivate this new method over case-incidence Rt methods. I think that including the AUC of the incidence-based method as a comparison on the main figures would be useful. I also think that an additional main figure showing comparisons of both methods and the true value for each scenario (over time) would be highly valuable. Currently the only version of this in the main text is Figure 1A (some in supplementary without CIs). The Ct-based Rt estimate in this figure seems very uncertain and noisy and looks like a far worse metric than the slightly delayed Rt estimated using case-incidence. For the future (or this paper), it would useful if your method could include some statistical smoothing so there is less noise in daily estimates, this would lead to more reliable inferred trends over time, particularly when sample sizes are small.

Given the relationship between Ct value distribution and incidence-based Rt estimates are used to parameterize the Ct-based model, can the Ct-based model ever perform better than using case data? Have you considered investigating model performance when you already have data quantifying viral shedding duration (so you don’t need to do a training period)?

Additional analyses highlighting the value of this method and detailing when it performs better or worse than incidence-based methods can provide valuable details on when it is useful to implement/what policymakers should weight the estimates from each method during different periods. One potential analysis that could demonstrate the value of this methodology is when case testing gets capped following the pandemic (due to limited resources). Your model could still be fit to incidence-based methods during the first wave, and then could be shown to be effective at estimating Rt following the reduction in testing capacity, when incidence-based methods would be highly biased. This is similar to one of the scenarios that you have already performed analyses for (this could be extended to consider different testing rate caps/sample sizes)

Minor comments:

Abstract

“However, it remains unclear whether this approach can be broadly applied to other pathogens, sources of virologic test data, or surveillance strategies beyond those used during the pandemic, such as in Hong Kong”: I was confused by the “, such as in Hong Kong”

“Influenza A” which one? Throughout the paper the authors refer to influenza A. It would be useful to know if it is influenza A H3N2 or H1N1 and If not some comment on if they have similar viral shedding patterns.

The last paragraph of abstract could be rewritten to emphasize the findings of this work. Currently the final paragraph doesn’t mention that the models were in general accurate. It only mentions that some models exhibited lower accuracy. This will likely need to be edited based on findings of model performance between incidence-based and Ct-based methods (see major comment)

Figure 1A – CT-based Rt seems all over the place compared to case incidence-based Rt? (see major comment)

Figure 3B – It would be useful to have more informative labels for “Pathogen X” so that the reader doesn’t have to refer back to Figure 1

Lines 104-160 read as Methods to me. I personally think this should be a methods section, but do not mind if this doesn’t change. The later methods section could be a supplementary methods section or moved up to join this section.

Discussion:

Lines 243-245. This links back to my major comment on the manuscript. The final sentence of this paragraph is not motivated by the findings of the paper. There is currently no description as to why this method would be used instead of incidence-based Rt estimation. This is particularly true given the relationship between Ct value distribution and incidence-based Rt estimates are used to parameterize the Ct-based model.

Lines 275-277. Very interesting ideas for future applications. Given the model here used case incidence-based Rt estimates to infer the relationship between Rt and Ct would additional data be needed for this extension?

Methods

Line 318 “symptomatic cases were detected at a fixed probability of 25%”. I assume this means that daily cases were pulled from a Binomial distribution ~B(N=infections, p=0.25)? Perhaps this could be stated for added clarity.

Line 390. The language is a bit unclear to me on how training periods were selected? It is more clear in the supplementary methods so maybe some language there can be added here.

Supplementary methods:

Estimation of incidence-based Rt: In estimating Rt using incidence-based methods you have assumed that delay distributions and incubation period were known exactly. In future work you could consider that these introduce biases that Ct-based methods could overcome (if you don’t train the model on incidence based methods).

Reviewer #2: This study evaluated the utility of cycle threshold (Ct) value-based estimation of the reproductive number (Rt) under varying surveillance conditions, epidemiological scenarios, and viral shedding kinetics. The method demonstrated robustness to variation in case ascertainment, provided that ascertainment rates remained stable over time and were not biased by disease severity. In terms of viral kinetics, accurate estimation was achieved when the viral load peaked before symptom onset, resulting in a monotonic shedding pattern that more reliably corresponds to time since infection. These findings are largely intuitive: stable detection minimally affects incidence-based Rt estimation, and a monotonic relationship between viral load and infection time allows Ct values to serve as a temporal marker. The authors’ simulations comprehensively support these conclusions. However, the manuscript would benefit from a more detailed discussion of the simulation assumptions, particularly regarding the detection mechanisms and their impact on the representativeness of population-level Ct distributions.

Major Comments:

1. The manuscript highlights the robustness of Ct-based Rt estimation under surveillance variability, but it would be helpful to see a more direct comparison with conventional incidence-based methods. Incidence-based Rt can also be biased, especially under delayed or inconsistent reporting. Under what conditions does the Ct-based approach clearly outperform case-based methods? Clarifying this point would improve the practical guidance for choosing between methods in real-world settings.

2. The simulations assume that only symptomatic individuals are eligible for detection and Ct measurement, with fixed detection probabilities and delays. How sensitive is the accuracy of Ct-based Rt estimation to the assumed symptomatic ratio? Since asymptomatic cases are common and often undetected (and contribution to transmission and viral dynamics might be different), it would be useful to understand how their exclusion affects the generalizability and robustness of the method.

**Have the authors made all data and (if applicable) computational code underlying the findings in their manuscript fully available?**

Reviewer #1: **No: ** The authors state: "All simulation data generated in this study and all codes for analyses will be made available at the GitHub repository ("https://github.com/vanialin/Ct_Rt_generic)"

This URL returns a 404 not found. The authors should make the GitHub repo public and create a DOI using a service like zenodo etc.

Reviewer #2: Yes

PLOS authors have the option to publish the peer review history of their article (what does this mean? ). If published, this will include your full peer review and any attached files.

**Do you want your identity to be public for this peer review?** For information about this choice, including consent withdrawal, please see our Privacy Policy .

Reviewer #1: No

Reviewer #2: No

**Figure resubmission:**
---

## [Editor Report · Decision Letter 1]

15 Sep 2025

Dear Dr. Yang,

We are pleased to inform you that your manuscript 'Generalizing population RT-qPCR cycle threshold values-informed estimation of epidemiological dynamics: Impact of surveillance practices and pathogen variability' has been provisionally accepted for publication in PLOS Computational Biology.

Best regards,

James M McCaw, PhD

Academic Editor

PLOS Computational Biology

Benjamin Althouse

Section Editor

PLOS Computational Biology

---

## [Editor Report · Acceptance letter]

PCOMPBIOL-D-25-01075R1

Generalizing population RT-qPCR cycle threshold values-informed estimation of epidemiological dynamics: Impact of surveillance practices and pathogen variability

Dear Dr Yang,

I am pleased to inform you that your manuscript has been formally accepted for publication in PLOS Computational Biology. Your manuscript is now with our production department and you will be notified of the publication date in due course.

With kind regards,

Zsofia Freund
